# Glucocorticoid Resistance: Interference between the Glucocorticoid Receptor and the MAPK Signalling Pathways

**DOI:** 10.3390/ijms221810049

**Published:** 2021-09-17

**Authors:** Lisa M. Sevilla, Alba Jiménez-Panizo, Andrea Alegre-Martí, Eva Estébanez-Perpiñá, Carme Caelles, Paloma Pérez

**Affiliations:** 1Instituto de Biomedicina de Valencia (IBV)-CSIC, 46010 Valencia, Spain; lsevilla@ibv.csic.es; 2Department of Biochemistry and Molecular Biomedicine, Faculty of Biology, University of Barcelona (UB), 08028 Barcelona, Spain; alba.jimenez.panizo@gmail.com (A.J.-P.); andrea.alegre.marti@gmail.com (A.A.-M.); evaestebanez@ub.edu (E.E.-P.); 3Institute of Biomedicine, University of Barcelona (IBUB), 08028 Barcelona, Spain; ccaelles@ub.edu; 4Department of Biochemistry and Physiology, School of Pharmacy and Food Sciences, University of Barcelona (UB), 08028 Barcelona, Spain

**Keywords:** glucocorticoids, glucocorticoid receptor, resistance/sensitivity, mitogen-activated protein kinases (MAPKs), dual-specific phosphatases (DUSPs), chronic inflammation, leukaemia, therapeutics

## Abstract

Endogenous glucocorticoids (GCs) are steroid hormones that signal in virtually all cell types to modulate tissue homeostasis throughout life. Also, synthetic GC derivatives (pharmacological GCs) constitute the first-line treatment in many chronic inflammatory conditions with unquestionable therapeutic benefits despite the associated adverse effects. GC actions are principally mediated through the GC receptor (GR), a ligand-dependent transcription factor. Despite the ubiquitous expression of GR, imbalances in GC signalling affect tissues differently, and with variable degrees of severity through mechanisms that are not completely deciphered. Congenital or acquired GC hypersensitivity or resistance syndromes can impact responsiveness to endogenous or pharmacological GCs, causing disease or inadequate therapeutic outcomes, respectively. Acquired GC resistance is defined as loss of efficacy or desensitization over time, and arises as a consequence of chronic inflammation, affecting around 30% of GC-treated patients. It represents an important limitation in the management of chronic inflammatory diseases and cancer, and can be due to impairment of multiple mechanisms along the GC signalling pathway. Among them, activation of the mitogen-activated protein kinases (MAPKs) and/or alterations in expression of their regulators, the dual-specific phosphatases (DUSPs), have been identified as common mechanisms of GC resistance. While many of the anti-inflammatory actions of GCs rely on GR-mediated inhibition of MAPKs and/or induction of DUSPs, the GC anti-inflammatory capacity is decreased or lost in conditions of excessive MAPK activation, contributing to disease susceptibility in tissue- and disease- specific manners. Here, we discuss potential strategies to modulate GC responsiveness, with the dual goal of overcoming GC resistance and minimizing the onset and severity of unwanted adverse effects while maintaining therapeutic potential.

## 1. Introduction

Glucocorticoids (GCs) are steroid hormones that regulate the physiology of all mammalian tissues throughout life due to their diverse roles in development, growth, metabolism, and inflammation [1,2,3,4]. In response to physiological cues and stressors, the hypothalamic-pituitary-adrenal (HPA) axis coordinates the systemic production and secretion of GCs from the adrenal glands in a circadian and stress-related manner to maintain tissue homeostasis. In turn, GCs mediate negative feed-back inhibiting secretion of the hypothalamic corticotropin-releasing hormone (CRH) and the adrenocorticotropic hormone (ACTH), limiting GC production.

The homeostatic control exerted by the HPA axis fits with an inverted U-shaped dose-response curve, where the equilibrium is achieved in the central range of the curve (optimal) while both GC excess or deficiency occur on either side of the curve (suboptimal effects) [2,5,6]. The disruption of this central HPA axis, due to pathophysiological triggers such as chronic stress, inflammation, or by prolonged exogenous GC treatments, results in abnormal endogenous GC levels. These GC imbalances contribute to disease and disease susceptibility in tissue-specific manners and with variable degrees of severity, by mechanisms that are not completely deciphered. Besides the adrenal GC production, certain tissues such as the thymus, intestine, brain, and skin, express functional equivalents of the HPA axis that allow for GC synthesis de novo [7,8]. This local GC production is increasingly recognized as a mechanism mediating rapid and critical control of immune activation [9]. 

Synthetic GCs are commonly prescribed to treat chronic inflammatory conditions including respiratory, autoimmune, and cutaneous diseases, as well as cancers of the hematopoietic lineage, mostly leukaemias [10]. The increasing number of GC-based prescriptions (around 3% of EU population uses GCs annually) [11] represents a high economic burden for health care systems, with demands rising as geriatric population and chronic diseases increase. However, despite their efficacy, long-term treatments and/or high doses with GCs trigger adverse effects, impacting metabolism (obesity, diabetes, and osteoporosis), and increasing susceptibility to stress and infections [10,12]. Another important limitation in the management of inflammatory diseases is resistance to GCs due to the lack of response or loss of efficacy over time [6,13,14,15]. The degrees of GC insensitivity are variable and highly dependent on the disease; for instance, while almost all patients with chronic obstructive pulmonary disease (COPD) and sepsis experience GC resistance, percentages are variable in patients with rheumatoid arthritis (30%), primary acute lymphoblastic leukaemia (ALL) (10–30%), or asthma (4–10%) [6].

Both endogenous and pharmacological GCs act through a dual system formed by the corticosteroid receptors GC receptor (GR/NR3C1) and mineralocorticoid receptor (MR/NR3C2), structurally and functionally close members of the nuclear receptor (NR) subclass NR3C that, upon hormone binding, act as ligand-activated transcription factors (TFs) [16,17,18,19,20]. While GR is ubiquitously expressed and almost exclusively activated by GCs, MR expression pattern is more restricted and can bind GCs and the mineralocorticoid aldosterone with similar high affinity. In tissues where GR and MR co-express, the selective activation of these TFs is achieved by pre-receptor mechanisms that modulate the local availability of active GCs. The enzymes 11β-hydroxysteroid dehydrogenase (HSD11B) type I and II, which catalyse the interconversion between active (cortisol) and inactive (cortisone) forms, represent a major regulatory mechanism for receptor selectivity [19,21,22]. 

## 2. Structure and Function of the Glucocorticoid Receptor (GR)

GR is encoded by *NR3C1*, a unique gene comprising nine exons (Figure 1A) that can give rise to multiple isoforms through alternative splicing (Figure 1B) and use of alternative translational initiation sites [10,17,18,23,24]. Similar to all NRs, GR is composed of modular domains required for ligand binding, receptor dimerization and binding of hormone receptor complexes to GC response elements (GREs) in target genes [10,16,17,18,25] (Figure 1B). The N-terminal domain features the activation function (AF)-1, which is ligand-independent, and also mediates interactions with the transcriptional machinery and co-regulators. Most sites for post-translational modifications (PTM) such as phosphorylation, acetylation, ubiquitination, and SUMOylation, map within this domain, contributing to modulation of receptor function [23]. The DNA-binding domain (DBD) is highly conserved among NR family members, and contains two zinc finger motifs that allow GR to recognize and bind to GREs (Figure 1C) as well as a nuclear localization signal (NLS)1. The DBD is connected to the C-terminal region by a flexible and poorly-conserved hinge region that is also target of PTMs. The C-terminal region includes the ligand-binding domain (LBD) that features the coregulatory binding pocket or AF-2 and an additional NLS2 (Figure 1D–F). The AF-2 groove, lined by helices 3, 5 and 12, is completed upon ligand-binding to the encapsulated ligand-binding pocket (Figure 1G). The LBD uses the AF-2 pocket to establish interactions with the LXXLL motif (L, leucine; X, any amino acid), also called the NR box of co-activators.

GRα and GRβ are identical up to residue 727, then GRα diverges with an additional 50 residues and GRβ holds an additional 15 unrelated residues (Figure 1B). A single-residue insertion on the DBD results in GRγ. Furthermore, alternative splicing generates GR-A and GR-P in malignant cells, which lack N- and C-terminal halves of the LBD, respectively (Figure 1B). It has been reported that GR isoforms exhibit differential expression across tissues with distinct subcellular localization patterns that impact transcriptional activity [2,16]. In this review, and unless specified otherwise, we will refer to GRα, the classical and most studied receptor isoform capable of ligand binding, as GR. 

The oligomerization of GR is still a matter of intense debate [26,27]. It has been observed that several of the GR domains contribute to self-association critical for functionality. Both the DBD (Figure 1C) and LBD (Figure 1D–F) domains dimerize and their crystal structures have been solved [28]. Despite common acceptance of DBD dimerization, LBD oligomerization is still debated with several models based on experimental structural data using X-ray crystallography and complimentary in silico methods [29]. Additionally, recent data in living cells obtained using the number and brightness technique have shown that the oligomeric state of GR bound to DNA may exhibit a more complex scenario where a monomer to dimer to tetramer transition is likely to occur [30]. To date, no detailed structural information is available either by X-ray crystallography or electron microscopy (EM) for full-length GR. However, recent EM data reported for the related androgen and oestrogen receptors may allow us to speculate that GR full length structure may resemble that of AR considering the higher degree of conservation of residues [31]. Furthermore, EM data have recently been published depicting how chaperones recognize the LBD of GR [32].

Upon ligand binding, GR dissociates from cytoplasmic heterocomplexes that include chaperones (HSP90, HSP70, and p23) and immunophilins (FKBP51 and FKBP52), undergoes PTMs, and translocates to the nucleus to regulate gene expression [4,10]. The GC-induced response is normally terminated by the autologous downregulation of GR following protein phosphorylation and subsequent ubiquitination through a PEST motif (rich in proline [P], glutamic acid [E], serine [S], and threonine [T]) in its C-terminus [4,10]. In a given cell type, the selective interactions of GR with TFs, co-regulators and/or chromatin modifying proteins determine receptor functionality [18,33,34,35]. This in part explains why different cell types vary in GR genomic binding and transcriptomic profile despite the ubiquitous expression of GR [36,37].

It is well established that GR anti-inflammatory properties rely on both DNA-binding–dependent and –independent receptor functions [36,38,39]. GR activates transcription of anti-inflammatory genes such as glucocorticoid induced leucine zipper (*GILZ/TSC22D3*) by binding to GREs in regulatory regions, and represses expression of pro-inflammatory genes such as *Il1-β* by binding negative (n)GREs, preventing the assembly of an active transcription complex. GR binding has also been reported at inverted repeat (IR) nGREs, unrelated to classical GREs [39,40]. Also, the expression of inflammatory genes can be suppressed by the recruitment of histone deacetylases (HDAC) to acetylated GR complexes bound to GREs.

On the other hand, GR can regulate gene expression through protein–protein interactions (or tethering) to pro-inflammatory TFs (typically NF-κB or AP-1) bound to their respective DNA binding sites by a mechanism known as transrepression [4,38,39]. GR PTMs such as SUMOylation are required for GR-mediated inhibition of inflammatory genes via repression of IR nGRE genes as well as transrepression of NF-κB/AP-1-dependent transcription [10,18]. Besides their classic anti- inflammatory actions, GCs can also act as pro-inflammatory mediators, and even exert dual roles depending on the pathophysiological context [41]. Indeed, transcriptomic and cistromic studies have revealed that GR co-recruitment with key inflammatory TFs such as AP-1, NF-κB, or STATs, results in enhanced transcription at a subset of co-regulated targets [36]. Among the extensive crosstalk between GR and other signalling pathways, this review is focused on the mutual interference of GR with the mitogen-activated protein kinases (MAPKs). 

## 3. Mutual Interference between GR and Mitogen-Activated Protein Kinases (MAPKs)

MAPKs are serine-threonine protein kinases that act as the meeting point for multiple upstream signalling pathways representing the final step of a phosphorylation cascade known as the MAPK module. In response to pleiotropic signals, including those from inflammatory cytokines, MAPKs dissociate from the MAPK module and can translocate to the nucleus to target TFs such as AP-1 and NF-κB or phosphorylate downstream protein kinases or other substrates to regulate gene expression [42,43] (Figure 2). Therefore, these kinases are critical in regulating inflammation although they also play key roles in regulating cell growth, apoptosis, and differentiation. In mammals, the MAPK family comprises extracellular signal-regulated kinases (ERK), c-Jun N-terminal kinases (JNK), and p38 MAPKs [43].

The ERK subfamily includes five members, among which ERK1 and ERK2 are the most extensively studied, being critical effectors of cell proliferation in response to growth factors. JNK family includes up to ten isoforms generated by alternative splicing from three genes (*Jnk*1-3). Importantly, JNK-mediated phosphorylation of the AP-1 component c-JUN is critical for AP-1-dependent transcription. The subgroup of p38 MAPK includes four isoforms (α, β, γ, δ), with different degrees of involvement in inflammatory disorders. 

Within the MAPK module, MAPKs are activated by upstream specific kinases known as MAPK kinases, MKKs or MAP2Ks, through phosphorylation on tyrosine and threonine residues, which in turn are activated by MAPK kinase kinases, known as MKKKs or MAP3Ks, whose activation is triggered by several signals involving RAS GTPases. Specific combinations of scaffold proteins/MKKK/MKK/MAPK allow for specific cellular responses for diverse upstream events towards effector proteins. Each MKK isoform acts specifically onto downstream MAPKs. MKK1 and MKK2, MKK4 and MKK7, and MKK3, MKK4, and MKK6 activate ERK, JNK and p38 MAPK, respectively (Figure 2).

MAPK activities can be inhibited by dual-specificity phosphatases (DUSPs or MAPK phosphatases MKP). Among them, DUSP1/MKP1 has a key role as a crucial anti-inflammatory mediator and gatekeeper of the immune response by dephosphorylating and targeting preferentially, but not exclusively, JNK and p38 MAPK [44,45]. However, under certain settings, the JNK/c-JUN pathway can also transcriptionally up-regulate *DUSP1* to limit the expression of inflammatory genes, representing an important feedback regulatory loop [46,47,48,49,50] (Figure 2).

GR can inhibit MAPK function through direct and indirect mechanisms [51,52,53,54,55,56]. For instance, GC-activated GR antagonizes JNK by inhibiting its phosphorylation through interaction with MKK7 [57,58]. Also, GC-activated GR transcriptionally induces *DUSP1* [55], an effect that is further increased in the presence of pro-inflammatory cytokines, as a feedback mechanism that enhances MAPK inhibition [2,33,34,46] (Figure 2). In addition to triggering transcription, hormone-bound GR increases DUPS1 protein half-life [55]. However, other studies showed that the role of DUSP1 in GC-mediated MAPK inhibition is transient suggesting that additional effectors act at later time points [59]. For instance, GC-induced GILZ interacts with RAS/RAF1 blocking phosphorylation and inhibiting downstream activation of MKK and ERK [60] (Figure 2). Importantly, GR-MAPK interaction is mutual and, as it will be discussed below, GR is also targeted by MAPKs; indeed, excessive MAPK signalling can result in GC resistance.

## 4. Overview of the Mechanisms That Regulate Glucocorticoid (GC) Sensitivity

The sensitivity to GCs is influenced by multiple heterogeneous mechanisms whose impairment may result in GC hypersensitivity or resistance, which can be generalized or tissue-specific, congenital or acquired [2,5,6,61,62]. Generally speaking, the transient exposure to GCs, endogenous or pharmacological, is beneficial while the persistence of GCs is detrimental [63]. Paradoxically, the excess and deficiency of GCs often share common features due to the existence of multiple feedforward and feedback loops aimed to limit the imbalances of GC signalling, which can ultimately lead to adaptation or induce disease [35]. The disparity of responses to a common trigger is directly related to the cell type- and context-specific actions of GCs [35,37,38]. This section summarizes the mechanisms that influence GC sensitivity, including impaired production and availability of GCs; GR polymorphisms and mutations; alterations in the levels, isoforms, or turnover of GR; GR PTMs; and GR crosstalk with other signalling pathways including MAPKs (Figure 3).

### 4.1. GC Production and Availability

The synthesis and secretion of GCs is regulated by the central and local HPA axes [2,5,9], and dysregulation of these regulatory loops can result in GC excess or deficiency. Patients with Cushing syndrome present elevated endogenous systemic levels of cortisol due to pituitary ACTH-secreting or adrenal cortical ACTH-independent tumours [5,6]. The clinical features of Cushing’s patients include immune dysfunction, depression, muscle wasting, skin atrophy, and adverse metabolic effects such as increased glycaemia, insulin resistance, hypertension, and adiposity, with associated increased morbidity and mortality rates [64]. Novel therapeutic strategies are envisaged after testing the use of MKK1/2 inhibitor MKK162, which show effectiveness in a mouse model of corticotroph tumor causal of Cushing’s disease, reducing tumor growth and tumor-derived circulating levels of ACTH and, hence, corticosterone [65].

Endogenous GC production also increases during physiological aging and contributes to the development of diabetes, obesity, and hypertension [21,22]. Importantly, pharmacological inhibition of HSD11B1 was able to significantly reduce several of these age and GC-associated adverse metabolic effects in experimental models. Therefore, HSD11B1 targeting has become a promising therapeutic strategy to reverse or ameliorate the tissue-specific consequences of increased GCs [21]. 

Other patients present a rare GC hypersensitivity syndrome despite normal circulating cortisol levels, and can develop signs of metabolic syndrome either spontaneously or due to steroid treatments. This disease has been linked to different mechanisms that include elevated GR levels, augmented GR-dependent transcription, and one reported GR mutation [66].

### 4.2. GR Polymorphisms and Mutations

GR polymorphisms are commonly found in the general population and can result in GC hypersensitivity or resistance [2,5,6,61,62]. The most common GR polymorphisms ER22/23EK (rs6189 and rs6190), N363S (rs6195), and BcII (rs41423247) have been linked with the risk of developing obesity, metabolic syndrome, and cardiovascular problems.

The inactivating heterozygous mutations of the *NR3C1/GR* gene are the cause of congenital primary generalized GC resistance or Chrousos´ syndrome (OMIM #615962), a rare genetic condition characterized by hypercortisolism without signs of Cushing’s syndrome. Due to the lack of the negative feedback loop on the HPA axis, patients also feature excess aldosterone and androgens, which result in hypertension, hypokalaemia and in women hirsutism and menstrual abnormalities. The overall impairment of GR-mediated signalling is highly variable in these patients, ranging from complete inability to respond to GCs to partially attenuated responses, and is also highly variable among tissues [2,5,62]. Most reported *NR3C1* mutations (approximately 30 cases) map at the GR-LBD, and only a few are located at the DBD [5,12]. In general, GR mutations result in decreased affinity for ligand, reduced nuclear translocation and/or DNA binding, and impaired interactions with coactivators such as GRIP [5,12,23,67].

It has been suggested that destabilization of the GR-LBD is the most common conformational change causing GC resistance; however, studies have mostly relied on computer simulations [68]. As the GR-LBD is implicated in oligomerization it is important to decipher the functional association between different GR oligomers and their pathophysiological function in Chrousos’ syndrome. Residues implicated in GC resistance are indicated in Figure 1C, E–G. As functional characterization of the mutants has been thoroughly revised [5,12,23,67], it will not be discussed in detail in this chapter.

### 4.3. GR Levels, Isoforms, Protein Turnover, and Post-Translational Modifications

Reductions in the pool of available GR affect hormone sensitivity, and can ultimately result in GC resistance [61]. The decreased expression of GR can be due to increased circulating endogenous GCs or chronic exposure to GC treatments. Also, the increased expression and/or altered ratio of the GR isoforms reported in several inflammatory diseases may be the origin of GC resistance [2,24]. The most characterized mechanism involves a dominant negative function of GRβ, unable to bind ligand, on GRα [24]. In fact, IL-1β imbalanced the GRα/GRβ ratio by increasing GRβ expression, resulting in a GC resistant phenotype that could be alleviated by JNK and p38 MAPK inhibitors [69]. The abnormal expression of other isoforms such as GRγ, GR-A, and GR-P has also been linked to GC resistance in patients with myeloma and leukaemia [2]. Imbalances in the relative levels of the GR chaperone complex proteins (high FKBP51 and low FKBP52) can also lead to GC resistance while the role of HSP90 and HSP70 in GC resistance is controversial.

GR is subjected to a variety of PTMs, including phosphorylation, acetylation, ubiquitination, SUMOylation, and oxidation, at evolutionally conserved sites (Figure 4). GR PTMs have a major impact on receptor function controlling GR half-life, subcellular location, and also modulate interactions with other signalling pathways that eventually affect GR-dependent gene expression [10,23,46].

GR is substrate of multiple protein kinases including cycling-dependent kinases (CDKs), AKT, glycogen synthase kinase (GSK)-3β as well as all MAPKs (Figure 4). While it is accepted that GR phosphorylation is crucial for its functional impact, the in vivo relevance at some sites is still controversial. In addition to cell-type specificity clues, reported differences might be due to the requirement of multiple GR PTMs to drive functional specificity. For example, GR phosphorylation has a prominent role in receptor turnover mediated by K419 ubiquitination [70,71].

Upon hormone binding, p38 MAPK and CDKs induce GR phosphorylation at S211, an event regarded as a hallmark for GR transactivation; this effect is further enhanced by phosphorylation at S203 by ERK and CDK (Figure 4). In addition, similar to AKT, p38 MAPK can also inhibit GR function by phosphorylating S134, which results in impaired GR nuclear translocation and/or transcriptional activation of a subset of GR target genes [72,73]. On the other hand, GR phosphorylation at S226, mediated by JNK, induces nuclear to cytoplasmic shuttling of the receptor, consistently inhibiting GR-dependent transcription [74,75] (Figure 4). Finally, GSK-3β-mediated phosphorylation at S404 alters the GR transcriptional program [76].

The acetylation of GR on K494 and K495 decreases the receptor´s transcriptional activity, and also blunts its ability to inhibit NF-κB. The reduction of GR anti-inflammatory capacity may lead to GC resistance in COPD patients which can be alleviated by HDAC2 overexpression [77]. Moreover, it has been shown that these acetylation sites are targeted by the circadian rhythm TF CLOCK/BMAL1 [78] and might mediate GC resistance in response to hyperglycemic conditions [79].

GR is also target of SUMOylation at K277, K293, and K703, with opposite outcomes; while modification at N-terminal sites has a negative effect on GR transcription that of K703 results in positive regulation [80,81]. Moreover, JNK phosphorylation on S226 promotes GR SUMOylation at N-terminal/inhibitory sites inhibiting GR transactivation [82]. Finally, there is evidence that, under oxidative conditions, C481 mediates suppression of GR functionality [83].

## 5. GC Resistance due to the Crosstalk between GR and MAPK Signalling

GR and MAPKs show strong interactions which normally result in mutual inhibition. In a given inflammatory context, the overall response to GCs is determined by multiple feedback and feedforward interactions between GR and cytokine-mediated signalling [13,14,15]. While many of the anti-inflammatory actions of GCs are achieved by the GR-mediated inhibition of MAPK activity, the GC anti-inflammatory capacity is decreased in conditions of excessive MAPK activation [13,14,15,33,46]. Given that chronic MAPK/AP-1-/NF-κB activation is a common denominator in multiple inflammatory diseases, the pharmacological inhibition of specific MAPK signalling pathway has become an add-on strategy intended to restore GC sensitivity [2,15,25]. As the interactions between GR and MAPK depend on the affected tissue(s) and are thus disease-dependent, the following sections have been organized according to the distinct pathologies associated to GC resistance.

### 5.1. Respiratory Diseases

Respiratory diseases such as asthma and COPD feature chronic inflammation of the airways. However, while asthma is characterized by allergic airway remodeling and hyper responsiveness with reversible airflow obstruction, COPD features chronic bronchitis, destruction of the peripheral airways and not fully reversible airflow limitation. Despite shared molecular mechanisms, these diseases differ greatly in their response to GC treatments. While most asthmatic patients have a relatively good response to GCs, almost all patients with COPD experience GC resistance [84]. It is accepted that the overexpression of pro-inflammatory cytokines such as tumor necrosis factor (TNF)-α, interleukin (IL)-17, IFN-γ, TGF-β, and IL-33, contributes to resistance in severe asthma and COPD, together with allergens, pathogens, and cigarette smoke. Among the mechanisms underlying impaired/lost GC sensitivity in these diseases, overactivation of the MAPK pathway and downstream AP-1-/NF-κB-dependent pro-inflammatory gene expression occupies a central position, as impairment of phosphorylation/dephosphorylation cascades ultimately results in defective GR binding, translocation, and function [13,52,85]. Consistently, several inhibitors targeting p38, JNK, and MKK1-2/ERK, such as SP600125, and trametinib, respectively, are in different stages of clinical development for treatment of respiratory diseases [86].

Asthma is a major health problem, with increasing incidence in western societies and the most common chronic disease in children [84]. The first-line therapy for patients with asthma with various degree of severity of all ages consists in inhaled corticosteroids, which achieve significant reductions in disease morbidity and mortality [13,84]. However, 4–10% of patients with severe asthma show poor responses to GCs, requiring higher doses to control the disease. About 1% of asthmatic patients are in need of treatment with oral GCs, which in addition to controlling asthma exacerbations, have undesired systemic effects [13,84].

In peripheral blood mononuclear cells (PBMCs) from severe asthmatic patients, GC-resistance correlates with higher levels of pro-inflammatory cytokines as well as increased expression and activity of the p38 α and β isoforms, relative to GC-responsive individuals [6,13,84]. The increased levels of cytokines in alveolar macrophages from asthmatic patients with reduced GC sensitivity result in the inhibition of GR function through its phosphorylation via p38α as well as the decreased induction of DUSP1 by GCs [13,87]. These findings are consistent with experimental models using macrophages from Dusp1 KO mice where the anti-inflammatory responses to GCs were reduced in vitro due to attenuated effects on gene expression and, as DUSP1 inactivates p38, as a consequence of increased p38 activation [88].

The anti-allergic actions of GCs are mediated through suppression of Th2-dependent pathways and involve the control of the TF GATA3 that regulates the transcription of Th2 cytokines such as IL4, IL-5, and IL-13. As both GATA3 and GR translocate to the nucleus via importin-α, GC treatments can inhibit nuclear translocation of GATA3 by competition and also by inducing DUSP1, which reverses the phosphorylation of GATA3 by p38 MAPK, necessary for its nuclear translocation [13,89]. These data highlight the importance of DUSP1 function to control p38 activity and thus modulate the sensitivity to GCs. Consistent with its causative role in steroid resistance, co-treatments with p38 inhibitors have proven useful in alveolar macrophages and PBMCs (SD-282 and SB 203580, respectively) isolated from severe asthmatics [84,90].

On the other hand, increased JNK activity due to the up-regulation of TNF-α and other pro-inflammatory cytokines has been reported in the airways of asthmatic patients, and associated to GC resistance in severe asthma. Augmented JNK1 activity results in increased phosphorylation at S226, resulting in decreased GR function [84]. These changes correlate with decreased expression of DUSP4/MKP2 in PBMCs from patients. The fact that enhancing DUSP4 activity restored corticosteroid sensitivity in vitro suggests that it might be a novel therapeutic target in severe asthma [84]. Corticosteroid resistance may also arise upon activation of PI3K and mammalian target of rapamycin (mTOR), leading to increased JNK1 activity [74,91]. Importantly, JNK inhibition (SP 600125) in mouse models for asthma had promising results in suppressing airway remodeling [84].

The activation of ERK by microbial antigens in severe non-allergic asthma also affects GR phosphorylation and can induce GC resistance in T cells in vitro [92]. ERK plays important roles in airway remodeling by regulating proliferation, extracellular matrix protein secretion, cytokine release, and eosinophil function in airway smooth muscle cells. ERK also promotes a Th2 phenotype and its inhibition blocks the production of Th2 cytokines in T lymphocytes. The use of MKK1/2 inhibitors to target the ERK pathway has proven effective for reduction of airway remodeling in several animal models of severe asthma. This is due to reductions in inflammation and in steroid resistance, as has been observed in airway type 2 innate lymphoid cells in severe asthma patients (trametinib) [85].

In COPD, airway inflammation results from the chronic inflammatory milieu in alveolar macrophages and the increased oxidative stress [84]. In PBMCs from COPD patients, basal and lipopolysaccharide-induced levels of IL-8 were higher relative to healthy smokers, along with higher increases in p38 activation. However, unlike in mild to moderate asthma, GCs are ineffective in reducing inflammation, which would rationally advise to limit this treatment to patients with severe disease in order to reduce exacerbations. Regardless, high doses of inhaled corticosteroids are routinely prescribed for patients diagnosed with COPD, increasing the risk of serious side metabolic effects such as diabetes and hypertension without proven benefits [84,90].

p38α also plays a central role in the pathobiology of COPD, and its activation seems critical for GC resistance. While p38 targeting in experimental mouse models of COPD was successful, the results of clinical trials assessing p38 inhibitors for COPD therapy have been so far disappointing [90,93]. Currently, the most promising strategy for the treatment of respiratory diseases such as asthma or COPD relies on the use of MAPK inhibitors as add-on therapies to inhaled corticosteroids or beta-blockers. A selective p38 inhibitor (GW856553) was reported to potentiate the repression of pro-inflammatory cytokines by GCs in PBMCs from COPD patients due to the decreased phosphorylation of GR-S211, mediated by p38 [87].

In addition, other therapies to restore GC sensitivity in COPD focus on upstream activators of the MAPK pathway. For instance, as PI3K/AKT overactivation in COPD can lead to reduced GR levels, PI3Kδ inhibition resulted in improved GC responses. Rapamycin, a immunosuppressant that inhibits the activity of mTOR, increased GC sensitivity in PBMCs from COPD patients through decreased phosphorylation of p70 S6 kinase, and concomitant reduction in c-JUN and cytokine (IL-8) levels. Phosphodiesterase 4 inhibitors potentiated the GC-induction of DUSP1 leading to reduction of p38 activity, and decreased IL-8 production [90]. Also, resistance to GCs in COPD patients can be mediated by a defect in GR deacetylation due to decreased HDAC2 expression in alveolar macrophages, which specifically inhibits NF-κB transpression by hormone-bound GR [77].

GCs also exert their anti-inflammatory effects by inhibiting the activity of phospholipases A2 (PLA2), which regulate the production of arachidonic acid, a precursor of lipid inflammatory mediators, playing important roles in many inflammatory diseases [94]. The reduced response to GCs in respiratory disorders such as acute lung injury may be due to reduced GR binding to cortisol and/or GRα overexpression, resulting in excessive activation of the MAPK cascade and insufficient regulation of downstream PLA2 pathway, therefore accentuating the pathologic response [95].

GC resistance has also been reported in other respiratory diseases such as chronic sinusitis. In an in vitro model, treatment with IL-1β resulted in the increased expression of GRβ relative to GRα, leading to induction of p38 and JNK activities [69].

### 5.2. Leukemias

GCs are extensively used as therapeutic agents for the treatment of almost all lymphoid malignancies, including leukaemias, lymphomas, and multiple myeloma, due to their immunosuppressive and anti-inflammatory properties. Studies using various cell lines derived from human hematological malignancies showed that ERK and JNK inhibitors may restore sensitivity to GCs [96].

GCs have been classically used for treating acute lymphoblastic leukaemias (ALL), haematological malignancies that originate from progenitors of B (B-ALL) or T (T-ALL) cell lineages that due to oncogenic transformation fail to differentiate and gain unlimited proliferative capacity. Initially, these malignant cells invade the bone marrow interfering with haematopoiesis. As the disease progresses they also colonize extramedullary sites including lymph nodes, spleen, and liver. Typically, ALL treatment starts with one-week of GC monotherapy followed by combination with other chemotherapeutic agents such as spindle inhibitors, genotoxic drugs, or antimetabolites [97]. The therapeutic response of ALL patients to GCs is highly heterogeneous. In paediatric patients, 10–30% of primary cases are poor- or non-responders, reaching up to 70% in relapsed cases [98]. As GC resistance strongly correlates with poor prognosis, the outcome of GC treatments is the main basis to design therapeutic strategies [99].

In immune cells, GC anti-inflammatory and immunosuppressive properties are due to promotion of cell cycle arrest and apoptosis. The inhibition of cell growth is mediated by the induction of CDK inhibitors such as p21/CDKN1A and downregulation of proliferative effectors like c-Myc or CDK4. Also, GCs regulate the balance between pro- and anti-apoptotic effectors through the transcriptional regulation of genes of the B-cell lymphoma 2 (Bcl2) family [100]. Therefore, in addition to loss-of function mutations, polymorphisms, or epigenetic downregulation of the NR3C1 gene, GC resistance in leukaemias is frequently caused by alterations in other signaling pathways and downstream targets. Indeed, GC resistance in ALL is consistently associated with alterations in the regulation of apoptosis, involving abnormal expression Bcl2 family members, inactivation of the tumor suppressor TP53 or overexpression of its inhibitor, MDM2. It can also involve alterations in other signal transduction pathways including Notch, IL7R/JAK/STAT, PTEN/PI3K/AKT/mTOR and RAS/MAPKs. As the apoptotic-related mechanisms of GC resistance in immune cells have been recently reviewed [99], this section focuses on the GC resistance associated with impairment in the MAPK-related pathways.

Genome-wide transcriptomic analysis assessed the therapeutic response to GCs in ALL and identified several prednisolone-induced genes related to MAPK pathways in cell lines and clinical samples [101,102]. Consistently, while pharmacological inhibition of p38 MAPK protected leukemic cells from GC-induced apoptosis ERK or JNK inhibition had the opposite effect, correlating with GR phosphorylation on S211 [103]. However, genetic and pharmacological downregulation of MKK2 and MKK4 enhanced prednisolone-induced apoptosis in B-lineage leukemic cells through distinct mechanisms. While the effects of MKK2 knockdown were mediated by increased TP53 expression and, hence, enhanced sensitivity to all chemotherapeutic agents, MKK4 downregulation specifically augmented sensitivity to GCs by increasing GR levels. Moreover, ERK activation was observed in patient samples at relapse compared to the time of diagnosis. Consequently, in xenograft models treatment with trametinib reduced leukemic cell burden from relapse but not from diagnosis samples [104].

In addition to upstream components of the MAPK pathways, downstream effectors have also been related to GC response in leukemia. Transcriptomic analysis in the leukemic Pre-B ALL cell line 697 subjected to GC treatment showed an unexpected repression of the GC target gene DUSP1 with concomitant induction of JNK activity [105]. Moreover, while DUSP1 overexpression did not influence GC-induced apoptosis, its silencing increased it [106]. These results are consistent with those obtained in GC-resistant T-ALL cell lines showing that a low dose of anisomycin, an activator of JNK and p38 MAPK, effectively sensitizes them to dexamethasone-induced cell cycle arrest and apoptosis [107]. However, other studies using a subset of GC-resistant lines derived from B-lineage cells, including Pre-B 697, showed that JNK inhibition re-sensitizes to GCs [108]. Therefore, further studies in primary and relapsed clinical samples are required to define the use of specific MAPK inhibitors to either enhance GC therapeutic response or alleviate GR resistance.

### 5.3. Skin Diseases

Despite the overall efficacy of topical GC treatments to combat prevalent inflammatory skin diseases such as psoriasis or atopic dermatitis [109], the response to GCs may be reduced or lost over time in some patients [6,110]. This phenomenon known as tachyphylaxis has been reported in human and mouse skin, and is defined by loss of the anti-proliferative actions of GCs on epidermal keratinocytes, vascular and lymphoid cells. Steroid tachyphylaxis is difficult to quantify as decreased effectiveness of GC treatments may be also due to the lack of adherence to prescription [111,112]. In mice, it was demonstrated that chronic exposure to GCs induced desensitization; instead of the initial rate of growth inhibition (around 10–15% of control level), keratinocyte proliferation returned to the basal levels and even increased after two weeks of treatment [113]. The activation of kallikrein (KLK)-related peptidase KLK6 may in part be responsible for the development of GC-desensitization [114]. KLK6 is induced by the phorbol ester PMA, a known inductor of MAPK, and is up-regulated in the skin of patients with several hyperproliferative/inflammatory diseases. Also, KLK6 is also induced upon chronic GC treatment likely as a mechanism aimed to regenerate the epidermis after GC-induced skin atrophy; in agreement with this, Klk6 KO mice showed decreased steroid tachyphylaxis [114].

Evidence from genetically modified mouse models demonstrate that the anti-proliferative, anti-inflammatory, and anti-tumor actions of topical GCs are mediated to a great extent through negative interference between epidermal GR and the MAPKs [109]. Upon topical application of PMA, mice with constitutive or adult-induced deletion of epidermal GR featured enhanced skin inflammation together with increases in p38 and ERK activities as well as in downstream expression of AP-1-target genes, relative to controls [40,115,116]. Also, when topical GCs were applied in combination with PMA, epidermal GR KO mice but not controls were partially resistant to the GC therapeutic actions [116]. The over-activation of MAPK/AP-1 activities was due to keratinocyte-autonomous actions upon GR loss, as shown in immortalized keratinocyte cell lines with GR deficiency, in which re-insertion of GR restored the altered MAPK activity to levels close to controls [117]. In addition, mice harboring a mutation that impairs GR SUMOylation (GR-K310 in mice/GR-K293 in humans) exhibited more severe responses to topical PMA and partial resistance to topically applied GCs highlighting the importance of this PTM for inhibition of NF-κB- and AP-1-transcriptional activities [118,119].

Increased MAPK activity has been observed in mouse and human psoriatic lesions and ERK, p38 and JNK are thought to play a pathogenic role in this disease [120,121,122]. Similar to PMA, repetitive treatment with imiquimod, a TLR7 agonist that induces psoriatic-lesions, increased the severity of skin lesions in GR epidermal KO mice vs. controls, correlating with higher activation of p38 and ERK activities [117]. Indeed, lesional and non-lesional skin of psoriatic patients featured decreased GR expression as well as reductions in the local de novo GC synthesis [123,124]. These findings imply that defective cutaneous GC signaling can not only aggravate dermatological conditions but also be an etiopathogenic cause for inflammatory skin diseases. The cutaneous HPA axis appears as key regulator and therefore, the rescue of local GC production is a feasible strategy to improve psoriasis [8,109,123,125,126,127].

Studies on skin carcinogenesis have been classically addressed by the two-stage chemical protocol in mice, which consists of topical application of an unique dose of the mutagen DMBA followed by repetitive treatments with the tumor promoter PMA. During tumor progression, skin changes correlated with the sequential activation of AKT, NF-κB, and ERK; ERK activity rose only during the later stages of malignant conversion of papilomas to squamous tumors [128]. GCs were able to inhibit mouse skin tumor progression only if applied at the beginning of the promotion stage. We demonstrated that the gain-of epidermal GR function resulted in resistance to chemically-induced skin tumors while its loss caused increased sensitivity to develop these tumors [129,130]. The anti-tumor role of GR was mediated through interference with AKT and NF-κB, upstream activators of MAPKs [129,130,131].

### 5.4. Autoimmune Diseases

Many autoimmune diseases such as rheumatoid arthritis and inflammatory bowel diseases (IBD) show reduced effectiveness to routine treatments with GCs [132,133].

Rheumatoid arthritis is a chronic systemic autoimmune disease, with a prevalence around 0.5–1% of the population, affecting mostly the elderly, in particular women. The disease affects the synovial joints, which are progressively damaged and destroyed causing disability, and is comorbid with cardiovascular pathologies. It is well established that MAPK activation and/or DUSP1 deficiency are involved in the pathogenesis of rheumatoid arthritis and may contribute to disease progression. GC treatment alleviates joint swelling, stiffness and pain, at least in part by modulating the MAPK/DUSP pathways, directly or indirectly [134]. Unfortunately, long-term treatments with GCs may lead to bone loss (or GC-induced osteoporosis).

Among the proteins involved in GC resistance, the pro-inflammatory protein macrophage migration inhibitory factor (MIF), which increases the production of pro-inflammatory cytokines and positively regulates MAPK activation, and GILZ, play major roles. The mechanism by which MIF increases MAPK phosphorylation involves the inhibition of DUSP1, thus counteracting the anti-inflammatory effects of GCs [135]. The overexpression of GILZ in endothelial cells reduced adhesion and inflammation by increasing the expression DUSP1 along with the inhibition of the TNF-induced activation of all MAPKs [136]. Importantly, MIF-mediated inhibition of DUSP1 requires GILZ, exemplifying how feedforward and feedback loops are responsible for modulating the sensitivity to GCs [135]. These multilevel regulatory mechanisms also highlight the importance of the MAPK/DUSP pathway, as the reduction of DUSP1—either due to elevated levels of MIF or GILZ deficiency—amplifies MAPK-mediated signaling.

IBDs such as ulcerative colitis and Crohn´s disease are chronic diseases associated with dysregulation of the immune response in the intestinal mucosa. GCs are prescribed as the main anti-inflammatory treatment in patients with moderate to severe disease. While approximately half of patients respond to GC therapy, around 30% show partial responses, and 20% are GC-resistant. Also, upon long-term treatment, around 20% of IBD patients become dependent, requiring GCs to maintain remission [133]. Current efforts focus on second-generation steroids that maximize the amount of locally available steroid in the intestine and minimize systemic bioavailability.

The mechanisms underlying GC resistance in IBDs include high levels of TNFα, IL-6, and IL-8, and low IL-10 levels, in steroid-resistant relative to sensitive patients, with activation of the MAPK/AP-1 and NF-κB pathways. MIF is also implicated in the pathogenesis of ulcerative colitis through activation of pro-inflammatory cytokines and subsequent anti-steroid effects [137,138]. As most cytokines are targets of major pro-inflammatory associated TFs, this scenario constitutes an auto-amplification loop for GC-resistance.

Besides their anti-inflammatory role, the therapeutic efficacy of GCs also relies on their capacity to restore the intestinal epithelial barrier function, which is impaired in IBDs. In Crohn’s disease, the mechanisms of defective barrier are related with dysregulation of tight junction components such as claudins-2 and -4. Claudins were regulated via DUSP1 and consistently, inhibition of DUSP1 but not p38 or MKK1/2 prevented the GC control of tight junction permeability in intestinal epithelial cells, suggesting substrates in addition to MAPK [139].

In patients with Crohn´s disease, it was demonstrated that steroid-sensitive and -resistant patients show different cell type-specific patterns of activation of pro-inflammatory pathways. GC response correlated with activation of p38, JNK1, and NF-κB mainly in lamina propria macrophages while GC-resistance was associated to activation of these mediators mostly in epithelial cells [140]. In line with these studies, treatment of patients with p38 and JNK inhibitor CNI-1493 achieved significant improvement of Crohn’s disease patients [141].

In ulcerative colitis patients, refraction to GC treatments correlated with higher levels of MIF in lamina propria macrophages, which resulted in p38 activation, and reduced inhibition of IL-8. Consistently, anti-MIF antibodies reduced p38 overactivation, ameliorating IL-8 production [142].

Given the major role of GCs in maintaining the equilibrium between pathogen clearance and tolerance mechanisms, abnormally high GCs can lead to excessive tolerance, contributing to chronic inflammation that may result in sepsis. Sepsis is a complex disease where generalized systemic inflammation and tissue damage lead to multiple organ failure and death. Sepsis is considered a global health priority with estimates around 30 million patients, and mortality rates around 30–50% annually [143]. In sepsis patients, the impaired inflammatory response transitions from excessive to immunosuppressive stages. There is no current treatment available other than antibiotics and fluids, and GC derivatives are routinely used as adjuvant therapy as almost all patients with sepsis undergo GC resistance.

During sepsis, MAPK activation features two phases, an early active phase of p38 up-regulation, followed by a second phase of ERK activation; both are suppressed by GR antagonists [144]. DUSP1 also exerts a main role on the efficacy of GC-mediated suppression of inflammation in sepsis, as shown by in vitro and in vivo evidences. Consistently, experiments in Dusp1^−/−^ mice mimicking the consequences of septic shock (caecal ligation and puncture and peritonitis) demonstrated refraction to GC treatments [45]. As GILZ is expressed in many tissues, being downregulated during acute inflammation and up-regulated at immunosuppressive stages, it appears as a potential candidate for escaping GC resistance in sepsis. Experiments in mice have shown that GILZ has therapeutic effects in models of sepsis, and indeed mice overexpressing GILZ were protected against lethal septic peritonitis [145].

As TNFα is a major inducer of GC resistance in sepsis, the recent demonstration that the TNFα/GR antagonism is mediated by reshaping the nuclear GR´s interactome constitutes a major finding that should allow for development of more specific targeting approaches [146,147]. In sepsis, GC resistance seems to be associated with high GRβ levels that result in inhibition of GRα-mediated gene expression.

## 6. Conclusions and Perspectives

The impact and extent of GC resistance is highly variable among individuals, and is also heterogeneous depending on the tissue, which constitutes an intrinsic difficulty associated to GC prescriptions. An ideal scenario would allow the determination of GC sensitivity prior to treatment, thus allowing for tailoring doses and regimes to patients ensuring an optimal personalized therapeutic treatment with an increased benefit vs. risk ratio.

The complexity of inflammatory networks and the multitude of signaling pathways involved in chronic inflammatory diseases justifies the efficacy of GC treatments as drugs that target all cell types by heterogeneous mechanisms. So far, targeting MAPKs represents a major therapeutic approach for chronic inflammatory diseases with GC resistance. Several p38 inhibitors for various uses have been patented, and multiple clinical trials have addressed the use of JNK and ERK inhibitors, mainly for hematologic malignancies. However, due to differences in the underlying mechanisms between inflammatory pathologies the development of an unique MAPK inhibitor that can be used as universal treatment is unfeasible. The failure or reduced efficacy of a given MAPK inhibitor is likely related to the intricate network formed by these kinases subjected to multiple feedback and feedforward loops that result in compensatory mechanisms. For instance, as p38 deficiency leads to increased activation of ERK and JNK, p38 targeting could lead to activation of other inflammatory cascades. Therefore, any therapy to restore GC sensitivity through MAPK inhibition would require careful adjustment of inhibitor dosage to keep compensatory mechanisms to a minimum. Once GC sensitivity is regained, GCs could be co-administered to sustain inhibition of inflammatory signaling.

Novel strategies should include targeting of alternative downstream regulators of the MAPKs, hopefully leading to minimize off-target inhibitory effects. Also, simultaneous targeting of more than one kinase along the cascade would potentially achieve synergistic effects.

## Figures and Tables

**Figure 1 ijms-22-10049-f001:**
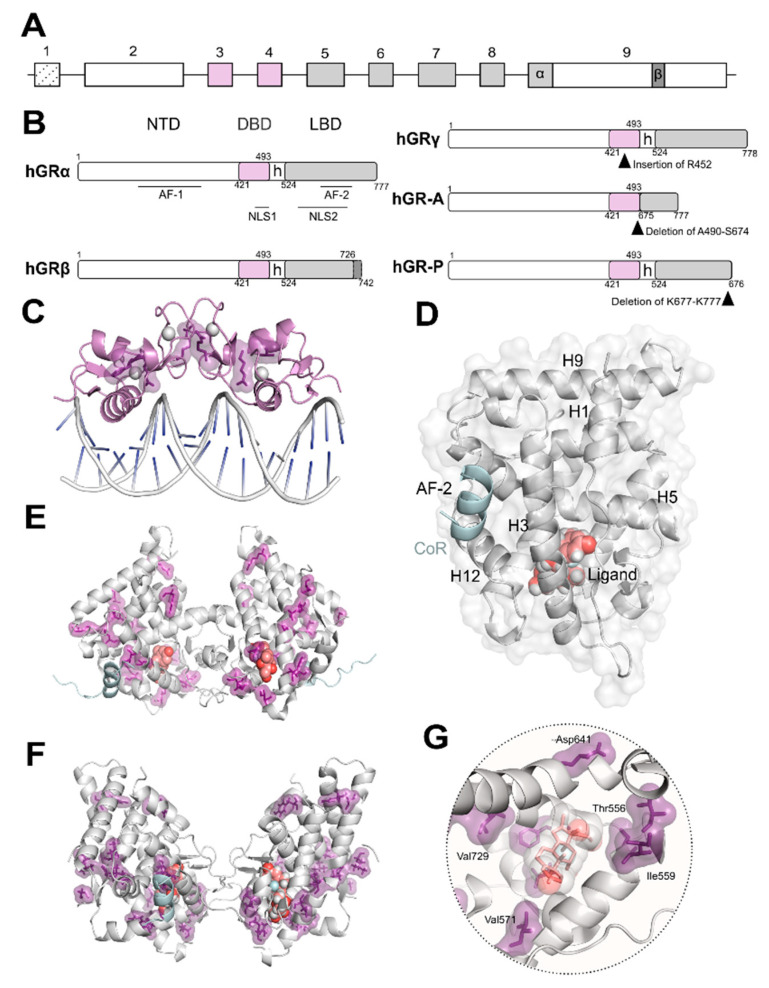
Glucocorticoid receptor (GR) isoforms: domain organization and crystal structures. (**A**) Schematic representation of *NR3C1/GR* gene organization. A unique gene gives rise to several isoforms through alternative splicing (**B**) and/or alternative transcription initiation. The exons encoding for the modular domains: N-terminal domain (NTD), DNA-binding domain (DBD), and ligand-binding domain (LBD), are indicated; exon 9 encodes the two alternative C-termini of the LBD in GRα and GRβ. (**B**) Schematic representation of GR isoforms domain organization generated by alternative splicing. hGRα modular domains include the intrinsically disordered NTD harboring the ligand-independent activation function (AF)-1; the DBD, the most conserved along the nuclear receptor superfamily; a poorly conserved hinge region (h); and the LBD displaying the surface-exposed co-regulator AF-2 groove. The five splice variants GRα, GRβ, GRγ, GR-A, and GR-P are shown; insertion/deletions are indicated by black arrowheads. (**C**) Crystal structure of the GR-DBD dimer bound to DNA shown in grey cartoon (PDB code 5CBX). This domain can dimerize on (+)GRE to activate transcription. Residues implicated in glucocorticoid (GC) resistance are shown (V423, R469, R477, F478) in purple as sticks and surface. The zinc ions are shown as grey spheres. (**D**) Cartoon representation of the three-dimensional secondary structure of the GR-LBD monomer (PDB code 5UFS) and surface. The domain is depicted in standard orientation (i.e., with helices (H)1 and H3 displayed in the forefront and the AF-2 pocket, lined by H3, H5 and H12 on the left hand-side). The ligand is shown as salmon spheres and the co-regulator peptide (CoR) is shown in cyan. (**E**) Crystal structure of the first GR-LBD dimer reported (PDB code 1M2Z). The ligand is shown as salmon-coloured spheres and CoR is shown in cyan. Residues implicated in GC resistance are highlighted in purple (T556, I559, V571, V575, L595, D641, Y660, L672, G679, R714, H726, V729, F737, I747, I757, L773). (**F**) Crystal structure of a GR-LBD dimer reported (PDB code 5UFS) displaying a different self-recognition pose than in (**E**). The ligand is shown as salmon spheres and CoR is shown in cyan. Residues implicated in GC resistance (T556, I559, V571, V575, L595, D641, Y660, L672, G679, R714, H726, V729, F737, I747, I757, L773) are highlighted in purple. (**G**) Close-up of the ligand-binding pocket of GR occupied by the ligand dexamethasone shown in grey spheres and salmon sticks. Residues implicated in GC resistance contacting the ligand (T556, I559, V571, V729, F737, I747, I757) and lining the LBP are shown in purple.

**Figure 2 ijms-22-10049-f002:**
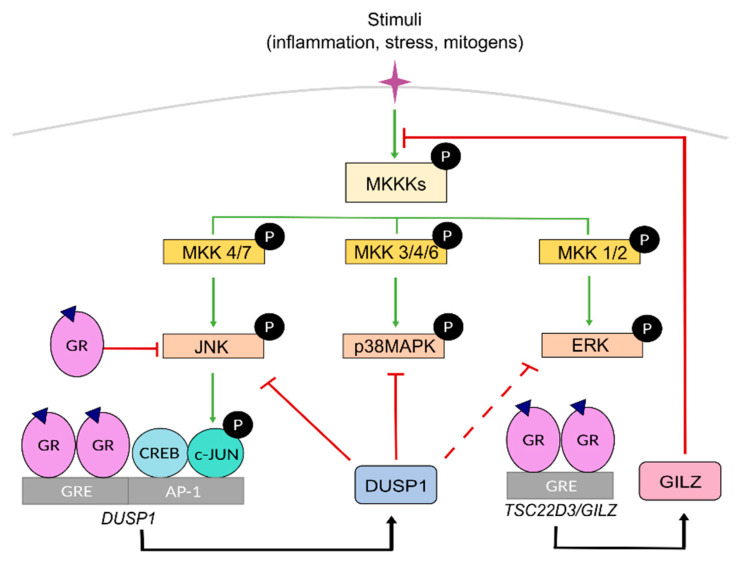
Mitogen-activated protein kinase (MAPK) signaling cascades and their inhibition by GR. MAPKs connect extracellular signaling with modulation of gene expression. In response to diverse stimuli, the kinases participating in the MAPK cascade phosphorylate and activate downstream kinases or other substrates including transcription factors. MAPK kinase kinases kinases (MKKKs) act on MAPK kinase kinases (MKKs) that specifically phosphorylate members of the MAPK JNK, p38, and ERK subfamilies which then signal downstream to regulate transcription of target genes dependent on AP-1 dimers (CREB/c-JUN) bound to AP-1 response elements. Ligand-bound GR negatively regulates MAPKs either by direct interference, as shown for JNK, or by regulating the expression of anti-inflammatory genes such as the MAPK phosphatase *DUSP1* or Glucocorticoid-induced leucine zipper (*GILZ/TSC22D3*). GILZ can also interact with proteins of the MAPK cascade to inhibit downstream activation of MKK1/2 and ERK. Green arrows and red blunt-ended lines indicate induction and inhibition, respectively. Dotted lines mark attenuated inhibition. Black arrows refer to the proteins encoded by GC-activated GR transcriptional targets *DUSP1* and *TSC22D3/GILZ*.

**Figure 3 ijms-22-10049-f003:**
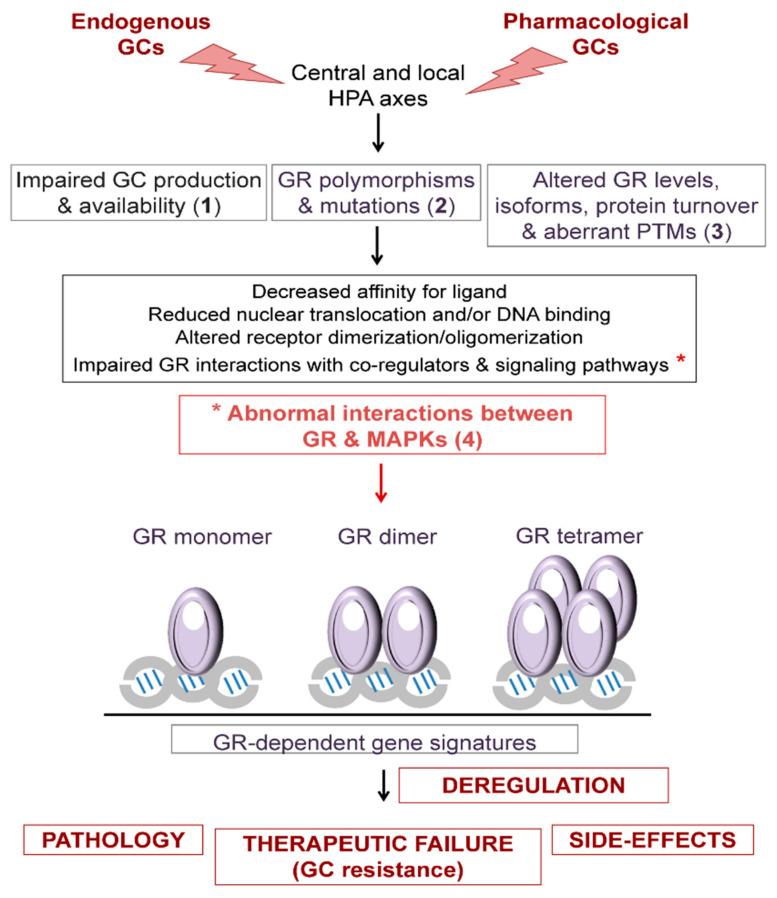
Mechanisms underlying GC sensitivity. Variations in the sensitivity to GCs can affect susceptibility to disease (endogenous) and to the capacity to respond to pharmacological doses of steroid analogs (exogenous). Impairments in either de novo production by the central and local hypothalamic-pituitary-adrenal (HPA) axes or in conversion from inactive to active GCs (availability) affect endogenous response and may lead to disease (1). GR polymorphisms and/or mutations (2), GR expression levels, protein stability, ratios of isoforms, and aberrant post-translational modifications (PTMs) (3) can reduce nuclear translocation and/or DNA binding, alter receptor dimerization/oligomerization, and affect GR interactions with co-regulators and signaling pathways. Among the abnormal interactions, those between GR and MAPKs can result in pathological changes to the GC response. Overall, defects in the mechanisms that influence GC sensitivity lead to deregulation of the GR-dependent transcriptional signatures.

**Figure 4 ijms-22-10049-f004:**
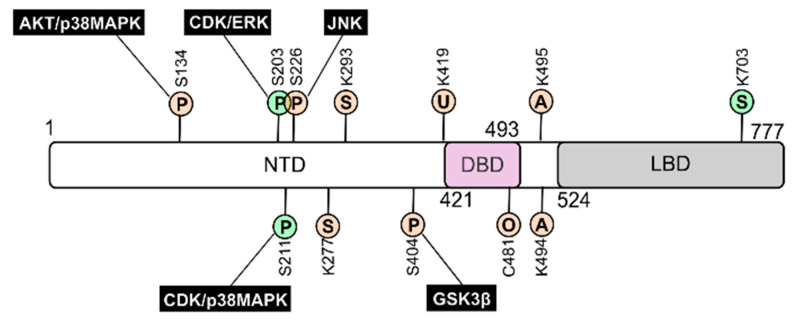
GR post-translational modifications: impact on receptor’s function. GR is subjected to a variety of posttranslational modifications (PTMs), which have a major impact on receptor function. Scheme indicates PTM target residues for hGRα including phosphorylation (P), acetylation (A), ubiquitination (U), SUMOylation (S), and oxidation (O). The major kinases phosphorylating GR and the outcome of the corresponding PTMs are also indicated (green, activation; salmon, inhibition).

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
