# Peer review of "Glucocorticoid Resistance: Interference between the Glucocorticoid Receptor and the MAPK Signalling Pathways"

_ijms, 2021, doi:10.3390/ijms221810049_

Round 1
Reviewer 1 Report
The review by Sevilla et al., entitled “Glucocorticoid Resistance: Interference between the Glucocorticoid Receptor and the MAPK Signaling Pathways”, discusses detailed mechanisms underlying glucocorticoid resistance syndromes and the impact of such resistance on responsiveness to endogenous or pharmacological glucocorticoid therapy. The review further captures the various potential strategies to modulate glucocorticoid responsiveness, aimed at overcoming glucocorticoid resistance and minimizing the onset and severity of unwanted adverse effects while maintaining therapeutic potential. This review is very relevant, especially considering the therapeutic utility of glucocorticoids in a wide array of diseases.
Minor Comment
Page 5; line 187 - The authors should check this subtitle and make it more straightforward.
Mitogen-Activated Protein Kinases (MAPKs). Mutual GR/MAPK interference
Author Response
- Reviewer: 1
We would like to thank the Reviewer for his/her positive comments. Following the reviewer´s suggestion, we have modified the subtitle in p. 5, line 187, which now reads: 3. Mutual interference between GR and Mitogen-Activated Protein Kinases (MAPKs).

Reviewer 2 Report
The authors review glucocorticoid resistance which focus on glucocorticoid receptor signalling through MAPKs pathways. The manuscript is well written and organised with reference on specific diseases.
However, I would like to address the following issues:
Regarding respiratory diseases, the interference of the enzymic isoforms of phospholipases A2 (PLA2) consists a whole chapter. Glucocorticoids are known in the literature to have a positive effect on the biosynthesis of lung surfactant and inhibitory effects on the inflammatory phospholipases A2 expression.
The review article would be more complete if addressed epigrammatically the following questions:
- Is there any bibliographic information on respiratory disorders as well as on other diseases regarding the effect of glucocorticoids on the expression of phospholipases A2 which are implicated in many inflammatory and malignant diseases ?
- Is this effect performed through GC —> GCR —> MAPKs —> PLA2 pathways ?
- Does glucocorticoid resistance affect the above route in some cases and in which way ?
Minor:
A short table list with all GCR isoforms would fit to the review.
Abbreviation for glucocorticoid receptor preferentially would be GCR instead of GR.
Line 84 and 273: unbold
Author Response
- Reviewer: 2
Comments and Suggestions for Authors
We would like to thank the Reviewer for his/her constructive comments. We agree that it is relevant to address the effect of glucocorticoids on the expression of phospholipases A2, in particular given their relevance in respiratory disorders and their connection with the MAPK pathway. Following the reviewer´s suggestion, we have included a sentence in p. 11-12, together with two new references: “GCs also exert their anti-inflammatory effects by inhibiting the activity of phospholipases A2 (PLA2), which regulate the production of arachidonic acid, a precursor of lipid inflammatory mediators, playing important roles in many inflammatory diseases [Pniewska & Pawliczak, 2013]. The reduced response to GCs in respiratory disorders such as acute lung injury may be due to reduced GR binding to cortisol and/or GRα overexpression, resulting in excessive activation of the MAPK cascade and insufficient regulation of downstream PLA2 pathway, therefore accentuating the pathologic response [Kitsiouli, Nakos, Lekka, 2009].”
Minor:
A short table list with all GCR isoforms would fit to the review.
This review refers almost totally to GRα, with brief references regarding the role of other GR isoforms in GC resistance. As Figure 1B includes a schematic representation of GR isoforms domain organization, and given the space limitations, we consider unnecessary to add this table.
Abbreviation for glucocorticoid receptor preferentially would be GCR instead of GR.
We are aware that both GR as well as GCR can be used as abbreviations for the Glucocorticoid Receptor. However, GR is most widely used in bibliographic references, and GCR is also used to refer to Glucocorticoid Receptor Resistance, which can be misleading in the context of this review. For these reasons, together with the use GR in all figures, we prefer to maintain GR as abbreviation.
Line 84 and 273: unbold
This has been amended.

Reviewer 3 Report
The review of glucocorticoid resistance is scientifically well written and the article is well structured.
Author Response
- Reviewer: 3
Comments and Suggestions for Authors
We acknowledge the Reviewer for his/her positive evaluation of our work.

Round 2
Reviewer 2 Report
The provided revised version and the answers to the addressed issues are adequate and the manuscript could be accepted to be
published in IJMS in the present form.